# Establishing Innovative Complex Services: Learning from the Active Together Cancer Prehabilitation and Rehabilitation Service

**DOI:** 10.3390/healthcare11233007

**Published:** 2023-11-21

**Authors:** Carol Keen, Gail Phillips, Michael Thelwell, Liam Humphreys, Laura Evans, Anna Myers, Gabriella Frith, Robert Copeland

**Affiliations:** 1Sheffield Teaching Hospitals NHS Foundation Trust, Sheffield S9 3TY, UK; laura.evans48@nhs.net; 2Advanced Wellbeing Research Centre, Sheffield Hallam University, Sheffield S9 3TU, UK; g.phillips@shu.ac.uk (G.P.); m.thelwell@shu.ac.uk (M.T.); l.humphreys@shu.ac.uk (L.H.); a.myers@shu.ac.uk (A.M.); g.frith@shu.ac.uk (G.F.); r.j.copeland@shu.ac.uk (R.C.)

**Keywords:** prehabilitation, cancer, service delivery, collaboration

## Abstract

Prehabilitation and rehabilitation will be essential services in an ageing population to support patients with cancer to live well through their life spans. Active Together is a novel evidence-based service embedded within existing healthcare pathways in an innovative collaboration between health, academic, and charity organisations. Designed to improve outcomes for cancer patients and reduce the demand on healthcare resources, it offers physical, nutritional, and psychological prehabilitation and rehabilitation support to patients undergoing cancer treatment. The service is underpinned by behaviour change theories and an individualised and personalised approach to care, addressing the health inequalities that might come about through age, poverty, ethnicity, or culture. Meeting the challenge of delivering high-quality services across multiple stakeholders, while addressing the complexity of patient need, has required skilled leadership, flexibility, and innovation. To support patients equally, regardless of geography or demographics, future services will need to be scaled regionally and be available in locations amenable to the populations they serve. To deliver these services across wide geographic regions, involving multiple providers and complex patient pathways, will require a systems approach. This means embracing and addressing the complexity of the contexts within which these services are delivered, to ensure efficient, high-quality provision of care, while supporting staff well-being and meeting the needs of patients.

## 1. Introduction

Cancer, and treatments for cancer, can have a devastating effect on individuals and those around them. Prehabilitation aims to improve the health of people with cancer before their treatment begins through exercise, nutrition, psychological support, and behavioural change interventions [1]. By empowering and supporting people with cancer to enhance their physical and mental health, prehabilitation can improve treatment outcomes, maximise resilience to the adverse effects of cancer treatment, and support people to live life as fully as they can after treatment [2,3,4,5]. Prehabilitation may, in some cases, allow people with cancer to access treatments that were not previously available to them [2]. Rehabilitation continues this support to people with cancer during and after treatment, thereby maximising their outcomes and minimising the consequences and adverse effects of cancer treatment. It can help people to get well and stay well, to become as independent as possible, and to minimise the impact on carers and support services [3].

Cancer is primarily a disease of ageing: in the UK, nearly two-thirds of cancer diagnoses occur in the over-65s and one-third in people aged 75, with over half of all cancer deaths occurring in people aged 75 and over. By 2020, there will be nearly 2 million people aged 65 and over with a cancer diagnosis [4]. Given an ageing population [5], the provision of effective cancer care to support patients to live well through their life spans will become increasingly important; cancer prehabilitation and rehabilitation can contribute to this. 

While the research evidence for cancer prehabilitation and rehabilitation is expanding, services that deliver these interventions at scale, within existing clinical pathways, are less common. Active Together is a new multi-modal (exercise, nutrition, and psychological care) cancer prehabilitation and rehabilitation service embedded within a large teaching hospital in the UK National Health Service. Its design, based on research-based guidelines [2,3], spans the rehabilitation continuum from preventive rehabilitation (prehabilitation), following diagnosis and maintenance rehabilitation (during treatment), to restorative and supportive rehabilitation after treatment.

This paper describes key learnings from the set-up and first year of running this service, including the challenges of operating in a multi-organisational collaboration, the complexity of the service, workforce demands, and expanding and scaling the service.

## 2. Setting

Active Together is based in Sheffield, a UK city with a population of 580,000. Sheffield sits within the region of South Yorkshire, which has approximately 1.4 million inhabitants. An area of high social deprivation, South Yorkshire has a higher incidence of cancer than the national average (622 new cancer diagnoses per 100,000 people per year, compared to 593) and higher cancer mortality rates (293 people per 10,000 per year, compared to 266) [6]. Responsibility for the overall provision of cancer care in the South Yorkshire region sits with the Integrated Care Board (ICB), which includes Sheffield plus three other municipal centres with secondary care facilities (Rotherham, Barnsley, and Doncaster) and community-based health care across a wider geography encompassing urban and rural areas. Provision of cancer prehabilitation and rehabilitation is included within the ICB’s strategy up to 2030 [7]. As a regional specialist centre, Sheffield provides cancer care to patients living in Sheffield and across the South Yorkshire region. Active Together, therefore, offers prehabilitation and rehabilitation to all patients receiving cancer care in Sheffield, regardless of where they live. Given the regional spread, some patients might live up to an hour by car away from the service centres, which can present obstacles to treatment such as cost or time to travel by car or public transport. 

Active Together was initially funded in 2021 for three years by Yorkshire Cancer Research, a regional UK charity that supports cancer research and services to improve the lives of people with cancer. Before this, there were no established cancer prehabilitation or rehabilitation services in the region. Active Together is delivered as a collaboration between the charity, Sheffield Hallam University’s Advanced Wellbeing Research Centre (AWRC), and the Sheffield Teaching Hospitals NHS Trust (STH), one of the UK’s biggest providers of integrated hospital and community-based health care. The Active Together service was co-produced by patients, clinical experts in the field, exercise specialists, commissioners, and academics over two years. The design utilised learning from previous studies of exercise and rehabilitation in cancer that examined current practice and factors that would be required to develop services and potential barriers to their implementation [8,9,10,11].

In the service design [11] (Figure 1), patients are referred to the service by their cancer-treating team following diagnosis and the decision to treat. The service is multi-modal, offering integrated physical activity and nutritional and psychological support to patients [2]. Prehabilitation, from initial assessment to the start of treatment, is focussed on supporting patients to become as fit (physically, nutritionally, and psychologically) as possible ahead of their upcoming treatment. Patients are then supported to maintain healthy behaviours to the best of their ability throughout treatment (chemotherapy, radiotherapy, surgery, and immunotherapy). Reassessment at the end of treatment establishes the current patient status, encompassing any consequences or side effects of cancer treatment. Through restorative rehabilitation, the emphasis is on recovery from treatment and working towards functional and lifestyle goals. The final phase supports patients to transition to self-management of health behaviours outside of the Active Together setting, e.g., at home or in local community services. 

Embedded within the service is an individualised and personalised approach to care. This is delivered in several ways, including stratification of patient needs and tailoring support accordingly [2], focus on patients’ priorities through adopting a “what matters to you” approach [12], and behaviour change techniques and strategies embedded throughout the service [13]. The service is currently delivered at the AWRC and two community facilities that combine health and exercise facilities, with plans to extend to more sites as the service expands. As a developing service committed to high-quality care, Active Together seeks and acts on feedback from patients through a regular patient forum and questionnaires, surveys, and performance data from staff, clinicians, and other stakeholders to review and modify the service design. This includes adapting delivery across the multi-modal approach to optimise patient care within the service. 

Alongside the service delivery of Active Together is an extensive evaluation [14]. This adopts a mixed-methods design, comprising an outcome and a process evaluation. The outcome evaluation will use a single group, longitudinal design to determine changes in quantitative outcomes in physical activity (6 min walk distance and hand grip strength), nutrition (body mass index and PG-SGA), and psychological well-being (PHQ-9 and GAD-7) of patients who attend the service. A comparative analysis of healthcare resource-use outcomes (e.g., hospital length of stay and readmissions) against historical patient data will also be conducted. The process evaluation will use service performance indicators, semi-structured interviews, and focus groups to explore the mechanisms of action and understand the contextual factors that influence delivery and outcomes. The integration of quantitative measures of psychological change mechanisms with outcome data might help to clarify the complex causal pathways within the service. Combining both an outcome and a process evaluation will ensure that data relating to the implementation of the Active Together service can be integrated into the analysis of outcome effectiveness measures. The data collection from this evaluation is due to be completed in June 2024, and the findings will be published subsequently.

The service is delivered by a single multi-disciplinary team of staff from the three collaborating organisations. The multi-modal clinical delivery comes from physiotherapists, exercise specialists, dietitians, and clinical psychologists, with academic staff from the AWRC leading the evaluation of the service and the charity team supporting patient communication and engagement. 

## 3. Findings

Active Together welcomed its first patient in February 2022 and received 350 referrals in its first year across three tumour groups: colorectal, upper gastro-intestinal, and lung. The service continues to expand, incorporating patients from additional tumour groups. While accepting referrals for any adult patient, the service supports a predominantly older population, with the mean age of patients referred to the service being 70 years. As described above, the full findings of the quantitative and qualitative evaluation of Active Together will be published in due course; however, early feedback from the services shows it to be well received by patients and clinicians. Patients are typically seen to make improvements across the multi-modal domains during prehabilitation and, while worsening through the treatment, can recover to pretreatment levels of physical, nutritional, or psychological well-being though rehabilitation, with some patients recovering to better health than before their cancer diagnosis.

Throughout setup and first year of service delivery, several challenges and learning opportunities presented themselves to the Active Together team. These are discussed here and summarised in Figure 2.

### 3.1. Multi-Organisational Collaboration

From the outset, staff from the three collaborating organisations have worked within a single team to deliver the Active Together service. This has allowed the service and its patients to benefit from the interdisciplinary skills and expertise across the three organisations [15]. For example, at an operational level, in developing patient information materials, the team has been able to draw on the knowledge and expertise of exercise specialists, the healthcare professionals’ clinical experience of working and communicating with patients, and the communications and marketing skills of the charity staff to develop a product that far exceeds that which could have been achieved by any of the groups working on their own. The collaboration also allows for streamlining and efficiencies, for example, in managerial and administrative roles.

Similarly at a strategic level, the combined innovation, achievement, and ambition of the three organisations has enabled Active Together to be established as an evidence-based, highly skilled service delivering excellent care to a diverse patient population with the greatest need. The collaborative partners continue to raise awareness of the service and its benefits locally, regionally, and nationally, aiming for expansion to continued healthcare funding for services across South Yorkshire. The capacity to think beyond a base knowledge domain and be curious and willing to learn, communicate, and collaborate with others has demonstrated value elsewhere [16] and is critical to delivering value across the cancer treatment pathway [17].

Driving healthcare service innovation by realising the synergistic opportunities from this collaborative endeavour has taken time, effort, reflection, and application of learning [18,19]. There has been a need to understand and accommodate the different organisational goals, cultures, and languages, to value the skills and expertise associated with different staff roles, and to appreciate where expertise sits within the team and where it does not. To achieve this has required high levels of flexibility in leadership styles and roles, as well as organisational design and processes and an openness and willingness to organise and address problems as they arise. This transformative approach to leadership and organisational governance is central to a re-imagined healthcare system, and previous calls have been made to ensure that leaders in the health system seek opportunities to leverage better outcomes by working across, and not just within, a system [20,21]. 

For innovation and collaboration to thrive, the people within the healthcare system—including patients—must feel empowered to contribute to the innovation process [18]. Therefore, the culture in which a service is located should nurture these behaviours, not constrain them. With this in mind, the Active Together service has been developed within the context of a multi-disciplinary academic research centre (i.e., the AWRC) driven by a shared purpose of delivering transformative change in health-related outcomes through collaborative endeavour. In this context, the AWRC has acted as a crucible, enabling different elements to come together and to interact, leading to the creation of something new. The value of service development outside traditional clinical settings is worthy of further exploration.

### 3.2. Complexity

Effective prehabilitation and rehabilitation services such as Active Together are complex. Patients are seen at multiple points by different professionals over an extended period as they move through treatment pathways that vary between tumour groups and individual patients. By their very nature, patients may have high levels of need due to their cancer and its treatment. Complexity is also driven by pre-existing comorbidities associated with this typically older patient population. To meet patient needs, the Active Together service has to liaise with multiple hospital services, e.g., surgery, oncology, anaesthetics, general practice, and other community healthcare services. Assisting patients in transitioning to self-management is also complex. It can require engagement with the many community-based leisure and support services across the region. This can be happening alongside the challenging transition of patients’ care from oncology to primary care teams [22].

The multi-modal intervention, addressing physical activity, diet, and psychological well-being, is complex in itself, and at its core is the intention to leverage the ‘teachable moment’ to help change patients’ long-term health behaviours [23]. This requires intentionality, is best achieved across multiple contact points, and requires practitioners to understand the individuals’ capabilities, opportunities, and motivations to engage [24]. 

Barriers to engagement and behavioural change can be influenced by demographic, social, economic, or cultural circumstances [25,26], which need to be considered within every patient interaction; these barriers drive inequality in patient outcomes. Active Together has taken steps to address these issues by ensuring that every eligible patient can use the service, allowing access for patients across age ranges and from different ethnic backgrounds. A written equalities impact assessment is in place, which is used to identify and regularly review challenges and actions in this field. There is a requirement for blanket referrals from cancer pathways to eliminate subconscious bias. Performance data are routinely analysed to ensure the index of deprivation of the patients who attend the service aligns with those who are referred, ensuring the service is not missing patients from areas of lower social deprivation. The service uses translation services to support access for patients who do not have English as a first language. Matters of geography and cost of travel are addressed by offering access to funded transport services and delivering exercise classes in a range of venues, specifically targeting locations where deprivation is higher. The most ‘in need’ patients, for example, those with complex health, physical, or learning disabilities, receive individual risk assessments and careful planning with other services involved in their care to accommodate their needs and ensure their access to the service.

The service has a focus on improving equality of access for older patients. Early analysis revealed that patients aged over 80 were the most likely to turn down or disengage from the service; therefore, steps were taken to address this. For example, a detailed analysis of barriers and enablers to older patients attending the service was undertaken, including consultation with members from the Active Together patient panel and a review of patient feedback questionnaires. As a result, the service has adopted additional measures that include focused behaviour change approaches to appointment booking, which reflect language and influences that will be familiar to this patient group. The frequency and timing of appointment reminders have been reviewed, and within resource limitations, the service offers a range of activities that will have an appeal across age ranges but specifically for older patients. The service is aware of the variation in digital skills in older age groups and is committed to a range of communication including digital, verbal, and written approaches. 

In meeting these challenges, Active Together has benefited from advanced service design [11] and significant experience in the delivery of complex rehabilitation services. Continuous review, innovation, and adaptation have been needed to optimise efficiency and meet the changing needs of the service as it has expanded [19]. Establishing appropriate clinical governance and maintaining thorough documentation ensure safety and efficiency against a complex background. Balancing staff well-being with the rapid expansion and change in delivery has been a challenge for all involved. Clear clinical and operational leadership has allowed service expansion while maintaining the quality of patient care [21].

While there are clear benefits from the multi-organisational collaboration within Active Together, it brings additional complexity. Delivering a high-quality service is the overarching priority for all three organisations, yet each has its own aspect on the sub-components of this and what “high quality” means to them in this context. Each organisation has its own financial, governance, and reporting requirements, which must all be met but which differ in format and in emphasis. While not insurmountable, there are challenges in managing a single team within which staff have different contracts, working cultures and arrangements, job titles, and career development expectations. 

The service has required skilled leadership to navigate the complexities of the collaboration and of the health system and organisations within which it sits. Shared leadership across the teams, disciplines, and organisations within Active Together systems has fostered the functioning of an integrated team, with shared responsibility and mutual influence [22]. Additionally, the Active Together team includes a consultant physiotherapist, bringing leadership across the four pillars of extended practice—leadership, expert practice, research, and learning—which, along with extensive support and guidance from senior healthcare and academic leadership, has enabled the service to navigate its complexity, as well as the wider systems transformation required [27,28].

### 3.3. Workforce

Cancer prehabilitation and rehabilitation is a relatively recent innovation, with research evidence evolving over the last 10 years [29]. Nationally and internationally, services delivering prehabilitation and rehabilitation are few in number, and those that exist are likely to have evolved from research trials or pilot studies [30]. Consideration is therefore required regarding the workforce delivering these novel services, which will need to develop and grow as services in this field expand. 

Evidence indicates the benefits of multi-modal services that support patients through the delivery of physical activity and exercise, nutritional support, and psychological support [2]. In Active Together, this has been delivered by a clinical team combining physiotherapists, exercise specialists, dietitians, and clinical psychologists. Physical activity and exercise support for patients is shared between physiotherapists and exercise specialists on the team based on risk assessment, with those higher-risk patients being supported by physiotherapists and lower-risk patients by exercise specialists. 

However, models of delivery vary as services have been adapted to fit local situations, with subsequent wide variation in the skills and professions employed across the multi-disciplinary team. Benchmarking and liaison with other UK services by the Active Together team has identified services that include combinations of physiotherapists, dietitians, exercise specialists, clinical psychologists, speech and language therapists, nurses, and occupational therapists [30]. 

Due to the novelty of cancer prehabilitation, the existing pool of staff across these professions with experience working in this area is currently small. The Active Together service has found that, where healthcare staff have experience working with cancer patients or in areas of rehabilitation, their experience is less often in cancer rehabilitation and exercise. Exercise specialists are likely to have less experience working with patients with the degree of complexity involved or to have had previous involvement in cancer care. Staff in management and leadership roles were initially required to draw on their experiences of other similar services as they implemented and expanded their knowledge base of delivering cancer rehabilitation services. Therefore, all staff across the cancer prehabilitation and rehabilitation workforce are likely to have individual development needs to address the difference between their previous experience and their new roles.

The integrated approach adopted by Active Together, with staff groups across professions and organisations working within a single team, is a new way of working for all those involved and has required staff to take time to understand each other’s roles and skill sets and to adopt a common language for effective communication. By working closely together, the staff groups have learned from each other to develop their knowledge and skills, resulting in the delivery of high-quality patient care, which is delivered by the team members best suited to meet the needs of the individual patient.

In such circumstances, highly selective recruitment is required to build a team with optimal levels of skills and knowledge spanning the team and a recognition that high levels of training will be required for new staff who join, as well as ongoing training and support for established team members. The Active Together team delivers this training primarily through sharing expertise across the team, using planned staff induction, internally delivered education sessions, staff shadowing, supervision, and clinical leadership. Additional skills are developed by accessing the expertise of clinical teams within the hospital trust, e.g., cancer nurse specialists who might share information about a specific tumour group or treatment modality and external training courses where indicated, e.g., masters-level training in prehabilitation or advanced level training in cancer and exercise rehabilitation for exercise specialists. Developing career pathways and capability frameworks can help guide this workforce development [31]. 

Effective prehabilitation and rehabilitation requires that patients modify or adopt new health or lifestyle behaviours, e.g., to be more active or eat more protein. Staff delivering services, therefore, need to be highly skilled in delivering behaviour change interventions, and such approaches need to be threaded through the entire service and considered in every patient interaction. Within Active Together, this has been achieved by training staff—including administrative and support staff—in motivational interviewing and behaviour change techniques and through a continuous learning approach, supporting staff to apply these techniques in their clinical care. Consideration of these approaches is also included in service design [11], as well as in approaches to patient information and engagement. 

Multi-modal services work most effectively where all staff have a good understanding and can offer support to patients with lower-level needs across each of the pillars of the service, as well as an ability to recognise patients with higher levels of need who will need support from their specialist colleagues. For example, all members of the Active Together team have received Level 2 psychological support training [32] and can offer support to patients presenting with non-complex psychological concerns while recognising those who are more at risk and require targeted interventions or support.

Further work is required to develop the cancer prehabilitation and rehabilitation workforce of the future locally, nationally, and internationally. This will require collaboration between healthcare providers and commissioners, education providers, and organisations in the leisure sector to identify, train, and develop the staff groups required as cancer prehabilitation and rehabilitation expands and require an openness to innovation of roles and across professional boundaries [33].

### 3.4. Scalability and Expansion

Sheffield is a specialist cancer centre within the South Yorkshire region. This requires patients from across the region to travel to Sheffield for their cancer treatment. However, this does not need to be the case for prehabilitation and rehabilitation. Indeed, patient engagement and adherence can be improved by providing services that are more geographically accessible, i.e., nearer to patients and with good travel connections [34]. To ensure equal care across the whole population of the South Yorkshire region, the intention is to expand Active Together such that every patient with a cancer diagnosis can access equitable, high-quality prehabilitation and rehabilitation services, without barriers based on geography. This will establish a regional prehabilitation service, with care continuing to be delivered in Sheffield, as well as in the three other municipal centres within the region. A central hub will support service delivery in the four centres through functions that will benefit from a unified regional approach such as learning and education, patient information, evaluation, and clinical leadership. In anticipation of this, understanding and engaging with key stakeholders at multiple organisational levels are important in scaling the service, as is the design of the organisational infrastructure of a regional service.

Expansion to regional delivery of the Active Together service will require consideration of a balance between standardisation and local adaptation. It will be important to have a significant degree of common practice across the regional service—this will avoid duplication in, for example, development of patient information, service design and adaptations, and staff training. This will maintain quality and allow for clear and easy communication with patients and stakeholders about the nature and content of Active Together. A standard outcomes framework will also enable evaluation across the regional service to determine programme impact as delivery is scaled. Standardisation will need to be coupled with a recognition that local context, e.g., geographic, demographic, healthcare pathways or workforce, will result in a need for services to adapt elements of programme delivery and infrastructure to ensure the ‘best fit’ with local resources, cultures, and patient needs.

It is anticipated that expanding the service while maintaining consistent quality and value will require careful planning to overcome the potential obstacles to achieving this aim. This could include new models of regional commissioning, a future ambition in the UK health system. Options for consideration would consist of potential sources for funding—public, private, and third sector—and the stakeholders who might be involved in the delivery of services. The geographic scope of services would need to be defined, as well as which patient groups would be included or excluded [30]. Active Together has established an existing model of care, including multi-modal prehabilitation and rehabilitation, and decisions would need to be taken regarding continuing or modifying this service as it goes forward [35].

Securing sustained funding for a regional service will depend on demonstrating the benefit of the existing service to patient outcomes and the system-wide impact that cancer prehabilitation and rehabilitation can have on the healthcare economy. Other similar services have shown that prehabilitation can reduce hospital length of stay, case complexity, and readmissions, with associated cost benefit [36,37,38,39], and a similar evaluation of Active Together is ongoing [14]. In addition, there are potential savings to primary care that prehabilitation and rehabilitation services might offer, such as reduced demand for general practitioner, district nurse, and falls or personal care services. 

Establishing the resources to deliver the service expansion will be challenging. The workforce considerations have been highlighted; these will need to be matched by the provision of physical resources within which to deliver the service. This is a community-based service, which can best be delivered in physical spaces that support physical activity, such as gyms, but that also allow for the clinical aspects of provision, e.g., dietetic or psychological well-being support. To meet the needs of the whole population and begin to address health inequalities that might come about through age, poverty, ethnicity, or culture, the locality of services also needs to be carefully considered in relation to the population it needs to serve. Such resources are scarce and in increasing demand from competing health services that see the value of working within appropriate community settings.

Capturing outcomes and learning from the experience of scaling Active Together to a regional service will be important in supporting the development of other similar regional or national services in future.

## 4. Conclusions

Multi-modal cancer prehabilitation and rehabilitation services will be essential parts of care for an ageing population. These novel services benefit from innovation in provision to optimise care for patients and value for health commissioners by improving patient outcomes in physical activity, nutrition, and health and in reducing cancer healthcare costs by, for example, reducing hospital length of stay. Implementation requires careful navigation of complexity and new ways of working to ensure efficient, high-quality provision while supporting staff well-being and meeting the needs of patients. The Active Together service provides early insight into how these services might need to be designed to allow access for patients from all demographics and geographies to have equal access to services. Further workforce development is required, which might come through interprofessional learning, development of shared competencies, and enhanced education provision to meet the needs of this developing workforce.

Cancer prehabilitation and rehabilitation demonstrate that scalable investment in modifiable lifestyle factors, coupled with behaviour change and personalised care, can benefit patients and wider healthcare systems. In future, these services should be available to all cancer patients. Furthermore, the principles underpinning them should be considered and integrated into a wider healthcare strategy and policy and adopted in other health conditions and systems.

## Figures and Tables

**Figure 1 healthcare-11-03007-f001:**
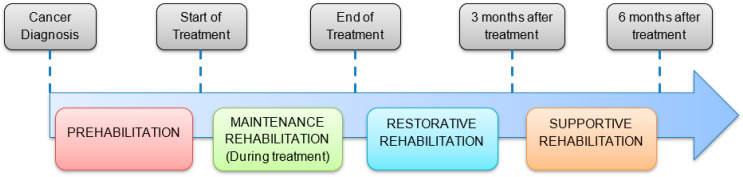
Active Together Design Overview.

**Figure 2 healthcare-11-03007-f002:**
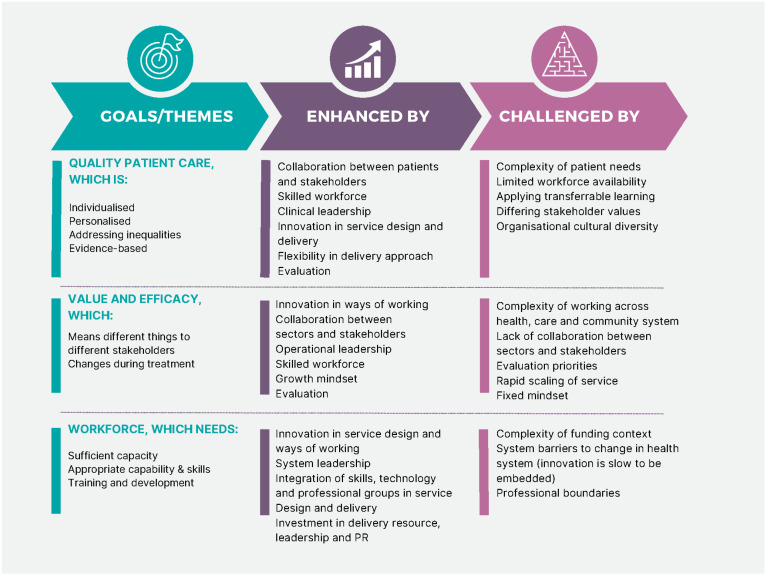
Summary of learning.

## Data Availability

Data are contained within the article.

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
