# Peer review of "Establishing Innovative Complex Services: Learning from the Active Together Cancer Prehabilitation and Rehabilitation Service"

_healthcare, 2023, doi:10.3390/healthcare11233007_

Round 1

Reviewer 1 Report

Comments and Suggestions for Authors

The abstract concisely overviews the challenges, importance, and considerations surrounding the "Active Together" service. It successfully communicates the relevance and urgency of the topic in the context of cancer care and prehabilitation/rehabilitation services. Given the constraints of an abstract, it would be beneficial to provide a bit more context on what the "Active Together" service entails. While the abstract describes the challenges and needs surrounding the service, a brief description of the service's components or main features could add value. The abstract might benefit from a slight rewording for clarity, particularly the "mooreand academia" segment, which seems like there might have been an error or omission.

Parte superior do formulário

The introduction provides a comprehensive understanding of the context and importance of prehabilitation and rehabilitation in cancer care. It covers the implications of cancer treatments, the benefits of prehabilitation and rehabilitation, the increasing aging population and its correlation with cancer, and the significance of the Active Together program.

The research design briefly describes the establishment and purpose of the Active Together service and the stakeholders involved in its creation. While it provides insight into the service's scope, the specific research methodologies and strategies used to gather data, analyze outcomes, and assess the program's efficacy are not profoundly elaborated. More detail about the research design, including any experimental or observational methods, data collection methods, sample sizes, and controls or comparisons, would help further assess the design's appropriateness. This section provides details on the setting and structure of the Active Together service, including its funding, collaborating partners, geographical reach, patient referral process, and the multi-disciplinary nature of its team. However, the specific methods used to evaluate the service, gather data, or the criteria for assessing patient outcomes are not articulated. As previously mentioned, more precisely delineating the research methods would be beneficial.

The results/findings are a detailed overview of the "Active Together" service, its challenges, successes, strategies, and workforce considerations. However, the structure feels more like a series of extended explanations and less like clear-cut results that might be expected from a research study or evaluation. The findings provide a comprehensive breakdown of older patients' challenges, how the program manages multi-organizational collaboration, and workforce training and development. However, the results of any evaluation or research regarding the efficacy or efficiency of the program are mentioned in passing and not explicitly detailed.

The reader is left with a good understanding of the Active Together program's structure and challenges but does not indicate its results regarding patient outcomes, cost benefits, or other measurable metrics. To improve, the "results" section could benefit from more precise delineation between different outcomes, incorporation of specific data points or quantitative metrics, and perhaps a summary that captures the overall impact and effectiveness of the program.

The conclusion provides a summary that reflects the challenges and innovations described in the earlier sections. It emphasizes the importance of cancer prehabilitation and rehabilitation in an ageing population and the significance of navigating complexities for optimal care. The statement about the importance of these services in an ageing population aligns with the earlier emphasis on older patients and their unique challenges and needs. The conclusion also acknowledges the need for new ways to ensure high-quality provision while considering staff well-being and patient needs, elaborated in the workforce and scalability sections. The reference to "Active Together service provides early insight" aligns with the earlier detailed exploration of the service's strategies, challenges, and accomplishments.

Reviewer 2 Report

Comments and Suggestions for Authors

Dear Authors,

I appreciate your submission of the manuscript “Establishing innovative complex services: learning from the Active Together cancer prehabilitation and rehabilitation service ”. Your research addresses an important topic in the field and contributes valuable insights.

After thoroughly reviewing your manuscript, I offer suggestions for improving your work’s quality, clarity, and impact. Please remember that these are recommendations, and you can decide how best to address them based on your understanding of the subject matter and your intended audience. Here are the recommendations:

Abstract:

Recommendations for Improvement:

    • Greater specificity: Certain statements can be made more specific to provide clarity on what exactly the service aims to achieve and how.
    • Refinement of scope: Instead of encompassing every challenge and solution, the abstract might benefit from a more focused scope.
    • Clarify terms and concepts: Ensure that terms used, like "careful navigation of complexity," are self-explanatory or explained in the article’s main body.
    • Highlight innovation: Given the title and the emphasis on an "innovative complex service," it might be beneficial to include a line or two highlighting what makes "Active Together" distinct from existing services.
    •  

Introduction 

Recommendations for Improvement:

    • Enhance Transition: Improve the flow between discussing the ageing population's implications and introducing the Active Together service.
    • Provide Context for Sheffield: A line explaining its significance or representativeness might add clarity if Sheffield’s data is being used.
    • Expand on "Evidence-based": Briefly mention what kind of evidence supports the Active Together service – is it based on previous pilot studies, established medical practices, or some other form of research?
    • Clarity on Paper's Focus: While the last line mentions that the paper will describe key learnings from setting up and running the service, it might help to briefly touch upon what aspects will be covered (e.g., challenges faced, patient outcomes, stakeholder involvement) to set reader expectations.

Setting

Recommendations for Improvement:

    • Detailed Figures: When mentioning deviations from the national average, actual figures or percentages help readers grasp the significance.
    • Logistical Challenges: Providing more details about the challenges faced due to patients living up to an hour away could emphasize service accessibility.
    • Co-production Insights: Elaborate a bit more on the co-production process, incorporating any challenges faced, or specific feedback that changed the service design.
    • Embed Figures: The design of "Active Together" is referenced as Figure 1, but without seeing this figure, it's challenging to comment on it. The text should ensure the figure complements and doesn't replicate the described information.
    • Diversity in Patient Base: The piece ends mentioning the service supports a predominantly older population. A brief mention of the service's adaptability to cater to varied age groups would be useful.
    • Feedback Loop: Detailing how ongoing feedback from patients and healthcare providers is being utilized to iterate and improve the service would be beneficial.

Findings

Areas for Improvement:

  • Quantitative Data: More quantitative data could be provided to measure the success and impact of the service. For example, of the 350 referrals, how many patients completed the program? What were the tangible outcomes?
  • Feedback Mechanism: While challenges are discussed, there's no mention of patient feedback. How do the patients feel about the service? Are there any qualitative testimonies or reviews that can be shared to provide a comprehensive view?
  • Comparative Analysis: It would be beneficial to see how "Active Together" compares to other similar services (if they exist). This would provide context on its effectiveness and areas where it might be outperforming or underperforming.
  • Organizational Overlap: With three collaborating organizations, there's bound to be some overlap in roles and responsibilities. This could be streamlined for efficiency. A clearer delineation of roles might also be beneficial.
  • Future Outlook: While the article focuses on challenges faced and how they were managed, it would be helpful to discuss forward-looking strategies. How does "Active Together" plan to scale, adapt, or improve in the coming years?
  • In-depth Examination of Demographic Challenges: The service seems to recognize demographic challenges, especially with older patients. Still, having an in-depth analysis of these challenges and more proactive solutions would be helpful.
  • Research and Efficacy: The section could benefit from more empirical data on the effectiveness of the multi-disciplinary approach. Do patients receiving care from more diverse teams have better outcomes?
  • Standardization vs. Adaptability: While adapting services to local situations is crucial, the article could delve deeper into how a balance between standardized protocols and adaptability is achieved.
  • Scalability: The challenges of scaling up the training and integrated approach in larger regions or nationally could be addressed. Can this model be replicated with the same efficacy?
  • Feedback Loop: It might be valuable to incorporate feedback mechanisms from both staff and patients to continually refine the model.
  • Cost Analysis: Given the novelty and potential costs associated with training, an analysis of the return on investment, both in terms of patient outcomes and financial metrics, would be informative.

 Conclusion

Suggestions for Improvement:

  • Incorporate Specifics: The conclusion could be strengthened by referencing specific aspects of the Active Together service that exemplify its success or areas of improvement.
  • Expand on Workforce Requirements: Given that the workforce is a significant part of the service's execution, the conclusion could benefit from a more in-depth analysis or mention of the specific workforce requirements and challenges.
  • Address the Broader Implications: The conclusion could address the broader implications of the Active Together service's findings. For instance, how might the insights from this service impact broader healthcare strategies or policies in the future?
  • Clarify the Meaning of "Equitable Access": The conclusion mentions providing "equitable access for patients," but it would be beneficial to specify what "equitable" means in this context. Does it refer to geographical, financial, or another form of accessibility?
  • Reinforce with Data: Including data points, even if briefly, could add weight to the conclusion. For instance, even if preliminary, mentioning any results from the Active Together service could bolster the statements made.
  • Provide Future Directions: While the conclusion summarizes the findings and insights, it might be strengthened by providing recommendations or future directions for cancer prehabilitation and rehabilitation services based on the Active Together insights.
Comments on the Quality of English Language

Areas for Improvement:

  • Sentence Length & Complexity: While the sentence structure is generally straightforward, some sentences are lengthy and contain information. Breaking them down could enhance readability.
  • Transition Phrases: While the content flows logically, incorporating more transition phrases could improve the smoothness of the narrative and better guide the reader through the article's progression.
  • Active vs. Passive Voice: The article uses the passive voice, which is common in scientific writing. However, occasional use of the active voice might make the text more engaging.

Round 2

Reviewer 1 Report

Comments and Suggestions for Authors

Thank you for your detailed response to my feedback and the amendments you have incorporated into the revised manuscript. I appreciate your efforts to address my concerns and provide further context to the "Active Together" service within the abstract. Your additional details have certainly enhanced the manuscript's clarity and comprehensiveness.

Regarding the research design, I understand that a comprehensive evaluation encompassing qualitative and quantitative methodologies is underway, with findings to be published in 2024. While this information is promising, the manuscript would benefit from a more immediate and clear exposition of the research methods applied within the current scope of the "Active Together" service. Even if preliminary, sharing the research design, data collection methods, and the criteria for assessing outcomes within the manuscript would contribute significantly to the understanding and assessment of the service's current efficacy and impact.

I noted a reference to a "C4" in your response concerning the structure of the findings section, which does not appear elsewhere in your letter. I believe there might have been an oversight. I encourage you to revisit my comment regarding the need for more precise delineation between different outcomes in the "results" section and the incorporation of specific data points or quantitative metrics that could offer a clearer view of the program's impact.

While I understand that comprehensive results will be available later, any preliminary data or observed trends that could provide early insight into the program's performance would be invaluable for readers and substantially strengthen the findings section.

In conclusion, I look forward to seeing these additional details incorporated into the findings section to convey a more transparent snapshot of the program's immediate effectiveness and impact. I believe these adjustments will further enhance the quality and credibility of the manuscript and provide the readership with a more robust understanding of the "Active Together" service.

Thank you once again for your commitment to improving the manuscript. 

Author Response

Thank you for your comments.  In response we have modified the text as shown below, providing further details on the data collection methods and outcomes  that will be used.  Without access to quantitative results from the evaluation until it is completed, we have described the early indications of outcomes from clinician and patients perspectives, with the "Findings" section of the paper focussed on learning from the practical challenges of setting up and delivering the service.

From line 115:

Alongside the service delivery of Active Together is an extensive evaluation.[6] This adopts a mixed-methods design, comprising an outcome and a process evaluation. The outcome evaluation will use a single group, longitudinal design to determine changes in quantitative outcomes in physical activity (6-minute walk distance, hand grip strength) nutrition (body mass index, PG-SGA) and psychological well-being (PHQ-9, GAD-7) of patients that attend the service.   A comparative analysis of healthcare resource use outcomes (e.g. hospital length of stay, readmissions) against historical patient data will also be conducted. The process evaluation will use service performance indicators, semi-structured interviews and focus groups to explore mechanisms of action and understand contextual factors that influence delivery and outcomes. The integration of quantitative measures of psychological change mechanisms with outcome data might help to clarify complex causal pathways within the service.  Combining both an outcome and a process evaluation will ensure that data relating to the implementation of the Active Together service can be integrated into the analysis of outcome effectiveness measures. The data collection from this evaluation is due to complete in June 2024 and the findings will be published subsequently.

From line 140:

The full findings of the quantitative and qualitative evaluation of Active Together will be published following completion of data collation in 2024, however early feedback from the services shows it to be well-received by patients and clinicians. Patients are typically seen to make improvements across the multi-modal domains during prehabilitation and, while worsening thought treatment, can recover to pre-treatment levels of physical, nutritional or psychological well-being though rehabilitation, with some patients recovering to better health than before their cancer treatment.

Reviewer 2 Report

Comments and Suggestions for Authors

Dear Authors,

I want to extend my heartfelt thanks for accepting my comments and considering them for revision in your manuscript. Thank you again, and I look forward to seeing the final version of your paper.

Author Response

No further changes are required from this reviewer